# Peer-led counselling with problem discussion therapy for adolescents living with HIV in Zimbabwe: A cluster-randomised trial

Victoria Simms[1]*, Helen A. Weiss[1], Silindweyinkosi Chinoda[2], Abigail Mutsinze[3], Sarah Bernays[4,5], Ruth Verhey[2], Carol Wogrin[3], Tsitsi Apollo[6], Owen Mugurungi[6], Dorcas Sithole[7], Dixon Chibanda[2,4,8‡], Nicola Willis[3‡]

1 MRC International Statistics and Epidemiology Group, London School of Hygiene & Tropical Medicine, London, United Kingdom, 2 Friendship Bench, Harare, Zimbabwe, 3 Africaid, Harare, Zimbabwe, 4 London School of Hygiene & Tropical Medicine, London, United Kingdom, 5 School of Public Health, University of Sydney, Sydney, Australia, 6 AIDS & TB Unit, Ministry of Health and Child Care, Harare, Zimbabwe, 7 Mental Health Services, Ministry of Health and Child Care, Harare, Zimbabwe, 8 Department of Psychiatry, University of Zimbabwe College of Health Sciences, Harare, Zimbabwe

‡ These authors are joint senior authors on this work.
* victoria.simms@lshtm.ac.uk

**Data Availability Statement:** All data files are available from the LSHTM Data Compass repository https://datacompass.lshtm.ac.uk/id/eprint/2142/.

## Abstract

### Background

Adolescents living with HIV have poor virological suppression and high prevalence of common mental disorders (CMDs). In Zimbabwe, the Zvandiri adolescent peer support programme is effective at improving virological suppression. We assessed the effect of training Zvandiri peer counsellors known as Community Adolescent Treatment Supporters (CATS) in problem-solving therapy (PST) on virological suppression and mental health outcomes.

### Methods and findings

Sixty clinics were randomised 1:1 to either normal Zvandiri peer counselling or a peer counsellor trained in PST. In January to March 2019, 842 adolescents aged 10 to 19 years and living with HIV who screened positive for CMDs were enrolled (375 (44.5%) male and 418 (49.6%) orphaned of at least one parent). The primary outcome was virological nonsuppression (viral load ≥1,000 copies/mL). Secondary outcomes were symptoms of CMDs measured with the Shona Symptom Questionnaire (SSQ ≥8) and depression measured with the Patient Health Questionnaire (PHQ-9 ≥10) and health utility score using the EQ-5D. The adjusted odds ratios (AORs) and 95% confidence intervals (CIs) were estimated using logistic regression adjusting for clinic-level clustering. Case reviews and focus group discussions were used to determine feasibility of intervention delivery.

At baseline, 35.1% of participants had virological nonsuppression and 70.3% had SSQ≥8. After 48 weeks, follow-up was 89.5% for viral load data and 90.9% for other outcomes. Virological nonsuppression decreased in both arms, but there was no evidence of an intervention effect (prevalence of nonsuppression 14.7% in the Zvandiri-PST arm versus 11.9% in the Zvandiri arm; AOR = 1.29; 95% CI 0.68, 2.48; *p* = 0.44). There was strong

**Funding:** DC and NW were awarded grant G-1710-02137 by the Children's Investment Fund Foundation (https://ciff.org/). VS and HAW are partly funded by the UK Medical Research Council (MRC) and the UK Foreign, Commonwealth and Development Office (FCDO) under the MRC/FCDO Concordat agreement and is also part of the EDCTP2 programme supported by the European Union. Grant Ref: MR/R010161/1 The funders had no role in study design, data collection and analysis, decision to publish, or preparation of the manuscript.

**Competing interests:** The authors have declared that no competing interests exist.

**Abbreviations:** AMD, adjusted mean difference; AOR, adjusted odds ratio; ART, antiretroviral therapy; CATS, Community Adolescent Treatment Supporters; CI, confidence interval; CMD, common mental disorder; CRT, cluster-randomised controlled trial; DBS, dried blood spot; FGD, focus group discussion; MoHCC, Ministry of Health and Child Care; PHQ, Patient Health Questionnaire; PST, problem-solving therapy; SMART, specific, measurable, achievable, realistic, and timely; SSQ, Shona Symptom Questionnaire; WHO, World Health Organisation.

evidence of an apparent effect on common mental health outcomes (SSQ $\geq$8: 2.4% versus 10.3% [AOR = 0.19; 95% CI 0.08, 0.46; $p$ < 0.001]; PHQ-9 $\geq$10: 2.9% versus 8.8% [AOR = 0.32; 95% CI 0.14, 0.78; $p$ = 0.01]). Prevalence of EQ-5D index score <1 was 27.6% versus 38.9% (AOR = 0.56; 95% CI 0.31, 1.03; $p$ = 0.06). Qualitative analyses found that CATS-observed participants had limited autonomy or ability to solve problems. In response, the CATS adapted the intervention to focus on empathic problem discussion to fit adolescents' age, capacity, and circumstances, which was beneficial. Limitations include that cost data were not available and that the mental health tools were validated in adult populations, not adolescents.

## Conclusions

PST training for CATS did not add to the benefit of peer support in reducing virological non-suppression but led to improved symptoms of CMD and depression compared to standard Zvandiri care among adolescents living with HIV in Zimbabwe. Active involvement of care-givers and strengthened referral structures could increase feasibility and effectiveness.

## Trial registration

Pan African Clinical Trials Registry PACTR201810756862405.

## Author summary

### Why was this study done?

- Common mental disorders (CMDs) such as anxiety and depression are highly prevalent among adolescents living with HIV. It is important to identify strategies to treat CMDs in this population.

- The Friendship Bench is a proven effective mental health intervention based on problem-solving therapy (PST), which is delivered by trained lay counsellors.

- The Zvandiri programme is a proven effective intervention to improve HIV outcomes among adolescents, delivered by trained peer counsellors.

- It is not known whether PST could improve mental health, and HIV outcomes, among adolescents living with HIV, when delivered in addition to the Zvandiri programme.

### What did the researchers do and find?

- We conducted a trial among 842 adolescents living with HIV in Zimbabwe, who also had CMDs (depression and anxiety), and attended public health clinics for HIV care.

- We randomly allocated 30 clinics to provide Zvandiri peer counselling to adolescents living with HIV, and a further 30 clinics to provide Zvandiri counselling plus the Friendship Bench PST.

- After a year, there was no difference in the proportion with unsuppressed HIV viral load, and this was low in both groups.

- There was a substantial improvement in mental health (depression and anxiety) in both groups, with significantly better outcomes among those in the Friendship Bench group.

- The peer counsellors adapted their training and focused on problem discussion rather than problem-solving, because many adolescents identified problems that they did not have the resources to solve.

## What do these findings mean?

- To our knowledge, this is the first study to show that an intervention can improve mental health among adolescents living with HIV who have mental health disorders.

- The lack of an impact on HIV viral load, compared to the Zvandiri programme, might be because of the effectiveness of the Zvandiri counselling and the presence of resistance to HIV drugs in a small number of participants.

- Mental healthcare should be integrated in HIV care for adolescents. It should be age specific, with shorter sessions than for adults, creating a space for discussing and sharing problems, and involving caregivers as appropriate.

## Introduction

Common mental disorders (CMDs) such as anxiety and depression are highly prevalent among adolescents living with HIV [1]. CMDs affect quality of life directly and are associated with impaired adherence to antiretroviral therapy (ART) and, therefore, with the increased resistance, morbidity, and mortality. Adolescents living with HIV have poorer virological suppression than any other age group [2].

The World Health Organisation (WHO) updated recommendations on service delivery for the treatment and care of people living with HIV [3] make a strong recommendation that psychosocial interventions should be provided to all adolescents and young adults living with HIV. One of the programmes underpinning this recommendation is Zvandiri, a WHO best practice programme [4] for adolescents living with HIV in Zimbabwe. The core of the Zvandiri approach is Community Adolescent Treatment Supporters (CATS). CATS are young people aged 18 to 24 years living with HIV who are trained and mentored to provide peer counselling and support. A cluster-randomised controlled trial (CRT) showed that the Zvandiri programme was more effective than standard of care at improving HIV virological suppression of adolescents but was not more effective for treating CMDs [5]. In qualitative interviews, participants reported that they found aspects of the intervention beneficial for mental health.

Friendship Bench is a counselling programme delivered by trained lay health workers, with a focus on problem-solving therapy (PST). PST is a cognitive-behavioural approach, which develops cognitive tools for problem solving, and builds adaptive skills and an enhanced sense of agency [6]. Friendship Bench was developed for adults and has been adapted for youth. In

adults, it has been proven effective at improving mental health outcomes compared to standard of care [6]. The Friendship Bench program focuses on exploration and understanding of the clients' situational context through talk therapy, mentalization, positive relational experience through being listened to, and intrapersonal growth towards strength and ability through goal-oriented learning.

The aim of the current trial was to evaluate whether enhancing the counselling skills of CATS to provide PST reduces virological nonsuppression and improves mental health among adolescents living with HIV in Zimbabwe, compared with standard Zvandiri care.

## Methods

The study design and methods have been fully described in a protocol paper [7]. Briefly, the Zvandiri programme was operational in 60 clinics (clusters), 6 in each of 10 districts across Zimbabwe. The clinics had previously been selected for a scale-up of the Zvandiri programme. In each district, the 6 clinics were randomly allocated 1:1 to the Zvandiri-PST arm or the Zvandiri arm by an independent statistician using a prewritten randomisation code. There was no allocation concealment.

All participants attending clinics allocated to the Zvandiri arm received Zvandiri standard care, consisting of HIV care following Ministry of Health and Child Care (MoHCC) guidelines [8], plus counselling and home visits from trained, mentored CATS, monthly support groups, and weekly text messages and home visits. Participants at the Zvandiri-PST clinics received the same, plus additional sessions based on the CATS' PST training. The CATS in the Zvandiri-PST arm met a Zvandiri mentor at least once every 2 weeks to review individual cases.

The PST consisted of a series of steps described in the Friendship Bench manual [9]. The first step is "kuvhura pfungwa" (opening up the mind), in which the client makes a list of all their problems. In the next step, "kusimudzira" (uplifting), the counsellor helps the client choose one manageable, relevant problem, establish a goal, and brainstorm solutions. The third step, "kusimbisa" (strengthening), focuses on selecting a detailed solution and devising a specific, measurable, achievable, realistic, and timely (SMART) action plan to carry it out. Finally, in the fourth step, "kusimbisisa" (further strengthening), clients are invited to join a support group.

Adolescents living with HIV aged 10 to 19 years who were taking ART were screened using the 14-item Shona Symptom Questionnaire (SSQ), a locally developed and validated instrument to assess symptoms of CMD [10,11]. Those scoring ≥7/14 who did not meet any of the exclusion criteria (unable to comprehend the nature of the study in either English, Shona, or Ndebele, currently in psychiatric care, end stage AIDS, current psychosis, intoxication, and/or cognitive disability) were enrolled after obtaining written consent from the caregiver and assent from the adolescent (or consent from the adolescent if aged 18 to 19). Those who were too unwell to participate or unable to give informed consent were excluded. The trial was registered with the Pan African Clinical Trials Registry (PACTR201810756862405) and approved by the ethics committees of the Medical Research Council of Zimbabwe and the London School of Hygiene & Tropical Medicine.

### Quantitative data collection and analysis

The primary outcome was the proportion of participants with virological failure (defined as ≥1,000 copies/ml) or death at 48 weeks after enrolment (plus or minus 8 weeks). Viral load was obtained from a dried blood spot (DBS) sample. Secondary outcomes were the proportion of participants with symptoms of CMD, defined as a score of ≥8/14 on the SSQ [11], and proportion with symptoms of depression, defined as a score of ≥10/27 on the Patient Health Questionnaire (PHQ-9) [11]. Poor quality of life was assessed as a secondary outcome using

the EQ-5D scale converted to an index using validated Zimbabwe utility scores [12,13] and analysed as a binary variable (1 versus <1), as the highly skewed distribution of scores did not allow for analysis as a continuous outcome. Severity of mental health symptoms was assessed using the SSQ and PHQ-9 as continuous scores.

The sample size of 840 participants recruited from 60 clusters provided 85% power to detect a difference in virological nonsuppression of 43% among participants in the Zvandiri arm versus 30% in the Zvandiri-PST arm assuming 20% loss to follow-up and a coefficient of variation (k) between clusters of 0.25. For secondary outcomes, the sample size provided 87% power to detect a difference in the proportion with CMD symptoms at 12 months of 16% in the Zvandiri arm and 8% in the Zvandiri-PST arm. The predicted outcomes in the Zvandiri arm were based on baseline results of a previous trial [5].

At baseline, data collection was predominantly paper based. A private company (Datalyst) completed double entry and validation of data. At endline, data were collected using a preprogrammed form in ODK on Android tablets. Data were exported to Stata 15.1 for cleaning and analysis, following a prespecified analytical plan. Statisticians were blinded to study arm until analysis was complete. Data were collected by CATS, so it was not possible to blind them to study arm.

Analysis used intention-to-treat principles, retaining participants in the arm to which they were randomised. In a prewritten analytical plan (S1 Text), an a priori decision was made to adjust for baseline values of the relevant outcome measure and for key variables that were deemed imbalanced between arms at baseline or were associated with missing outcome data. The primary analysis was complete case. For binary outcomes, logistic regression random effects models were used to estimate adjusted odds ratios (AORs) and 95% confidence interval (CI), with a random effects term to allow for clustering by clinic. A quadrature check was performed to evaluate the model fit. For continuous outcomes, analogous mixed effects linear regression models were used to estimate adjusted mean differences (AMDs) and 95% CI. Prespecified effect modification by age group at baseline of the intervention effect on the primary and secondary outcomes was assessed by fitting separate models for the 10 to 14 and 15 to 19 age groups.

## Qualitative data collection and analysis

To better understand how the CATS experienced implementing the intervention, including its feasibility and any necessary modifications, we collected qualitative data to capture concurrent and retrospective accounts of the CATS (Table 1). Case reviews between 20 individual CATS and their mentors were conducted each month over the 12-month trial duration. Two focus group discussions (FGDs) were held with 20 CATS at the end of the trial. Conducted by Zvandiri researchers at the Africaid offices in Harare, the FGDs involved a range of activities to

**Table 1. Description of qualitative data collection.**

| Method | Outline of method | Purpose of data collection |
|---|---|---|
| Case reviews | Individual supervisory discussions (between CATS and mentors) about client cases | CATS' experiences of provision of support and support needs in real time |
| FGDs | Group discussion between CATS | Retrospective reflections of CATS' on their experiences |
| Audio diaries | Recorded interviews between CATS and mentors about the CATS' experiences of delivering the intervention and participants' problems | Participatory insights into the types of problems participants presented with and the experiences of implementation |

CATS, Community Adolescent Treatment Supporters; FGD, focus group discussion.

facilitate reflective discussion. All data collection was conducted in the local languages and audio recorded. Data were transcribed and translated into English. A thematic analytical approach was adopted [14]. Analytical memos and weekly analytical team meetings were also used in the development of themes and identification of patterns related to CATS experiences across the datasets [15]. See S1 CONSORT Checklist.

## Results

### Quantitative findings

Between 2 January and 21 March 2019, 1,573 adolescents were screened for eligibility and 863 were eligible. The most common reason for noneligibility was an SSQ score below 7 (*N* = 690). In addition, 11 adolescents were excluded because they lived out of the study area, 6 because they were younger than 10 or older than 19 years, 2 because of disability, and 1 because they were at boarding school. Of the 863 eligible adolescents and their caregivers who were asked for consent or assent, 12 (1.3%) participants and 9 (1.0%) of caregivers refused. Of the 842 enrolled participants, 421 participants were randomised to each arm (Fig 1).

The mean number of participants per clinic was 14.0 (standard deviation 2.7, range 6 to 22). At baseline, 35.1% (292/833) of participants had a viral load ≥1,000 copies/ml, with 9 missing viral loads. The prevalence of virological nonsuppression by district ranged from 20.2% to 55.4%. There was little imbalance between arms in any of the prespecified variables (Table 2).

Endline interviews were completed between 4 to 31 January 2020, with the exception of one interview completed on 26 February 2020. Three participants died during the follow-up period, all in the Zvandiri-PST arm, of causes unrelated to the intervention. In addition, 22 participants moved away from the area, and 52 were lost to follow-up for unknown reasons. Follow-up was 89.6% in the Zvandiri-PST arm and 92.2% in the Zvandiri arm. The proportion lost to follow-up by district ranged from 2.4% to 16.7%. Fourteen participants had an endline interview but no endline viral load result, owing to challenges with lab testing as a result of the COVID-19 lockdown in Zimbabwe. Therefore, 88 participants (10.5%) had no primary outcome (excluding the 3 who died).

The mean time between baseline and follow-up interviews was 49.6 weeks in the Zvandiri-PST arm (range 44 to 56) and 49.4 weeks in the Zvandiri arm (range 42 to 56). All endline visits were within the prespecified visit window of 48 weeks ± 8 weeks. There was no evidence of an association between loss to follow-up and baseline values of virological nonsuppression, SSQ score or PHQ-9 score. Follow-up was similar by arm (89.1% versus 90.0% for the primary outcome; 89.6% versus 92.2% for other outcomes). Participants lost to follow-up were on average older than those with an endline measurement, and after adjusting for age, there was no association of loss to follow-up with any other factors (S1 Table). Therefore, we adjusted for baseline age in all analysis.

At endline, there was no evidence of an intervention effect on the primary outcome of virological nonsuppression or death (14.7% versus 11.9% in the Zvandiri-PST arm versus Zvandiri arm; AOR = 1.29; 95% CI 0.68, 2.48; *p* = 0.44) (Table 3). However, there was evidence of an apparent intervention effect on prevalence and severity of common mental health outcomes (SSQ ≥8: 2.4% versus 10.3%; AOR = 0.19; 95% CI 0.08, 0.46; *p* < 0.001; AMD = −1.14; 95% CI −1.80, −0.49; *p* = 0.001; PHQ-9 ≥10: 2.9% versus 8.8%; AOR = 0.32; 95% CI 0.14, 0.78; *p* = 0.01; AMD = −1.14; 95% CI −2.01, −0.27; *p* = 0.01) (Table 3). The proportion of participants with an EQ-5D score below 1 was lower in the Zvandiri-PST arm than in the Zvandiri arm, but with weak evidence of effect (27.6% versus 38.9%; AOR = 0.56; 95% CI 0.31, 1.03; *p* = 0.06). Crude results, unadjusted for baseline outcome or age, were similar (S2 Table).

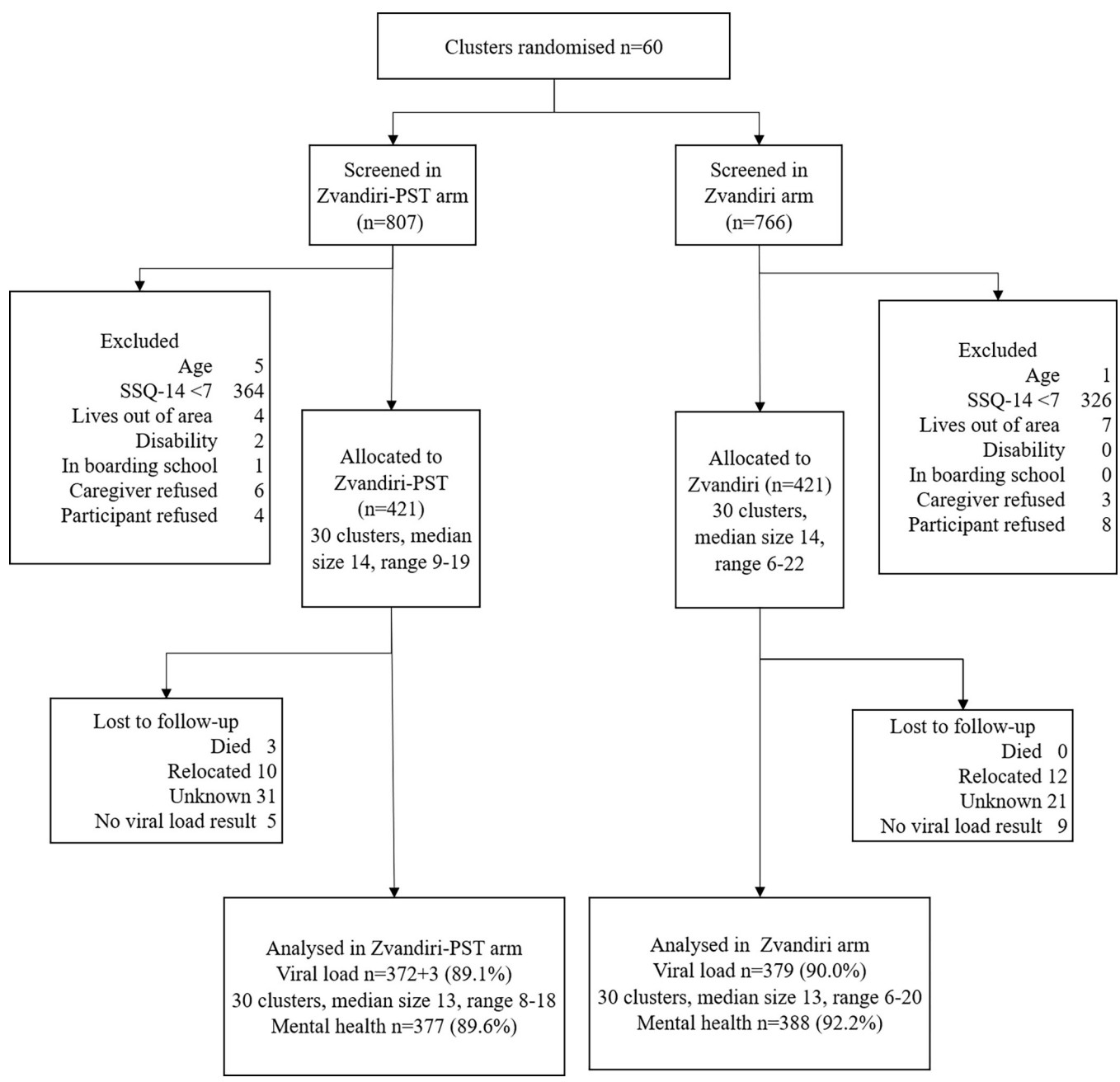

**Fig 1. Enrolment flow chart.** CATS, Community Adolescent Treatment Supporters; PST, problem-solving therapy; SSQ, Shona Symptom Questionnaire.

There was evidence of a difference in effectiveness of the intervention on virological non-suppression or death by age ($p = 0.01$ for interaction), with stronger evidence of an intervention effect in 15- to 19-year-olds (18.5% versus 9.8%; AOR = 2.65 95% CI 0.97, 7.23; $p = 0.06$) than for 10 to 14-year-olds (10.6% versus 13.6%; AOR = 0.68, 95% CI 0.35, 1.31; $p = 0.24$) (S3 Table). There was no evidence of effect modification by age for secondary outcomes. Intracluster correlation was below 0.2 for all outcomes.

Five participants in the Zvandiri-PST arm had no PST sessions logged, and 4 of these did not have endline data. The mean number of PST sessions was 5 (maximum 11) (Table 4). On

**Table 2. Descriptive characteristics of participants at baseline by trial arm.**

| | | Zvandiri-PST, n (%) | Zvandiri, n (%) |
|---|---|---|---|
| | N | *421* | *421* |
| Sex | Male | 184 (43.7) | 191 (45.4) |
| | Female | 237 (56.3) | 230 (54.6) |
| Age (years) | 10–11 | 77 (18.3) | 75 (17.8) |
| | 12–13 | 72 (17.1) | 92 (21.9) |
| | 14–15 | 78 (18.5) | 101 (24.0) |
| | 16–17 | 109 (25.9) | 68 (16.2) |
| | 18–19 | 85 (20.2) | 85 (20.2) |
| Education (*7 missing*) | Below grade 7 | 150 (36.1) | 151 (36.0) |
| | Grade 7 | 153 (36.8) | 133 (31.7) |
| | Secondary or higher | 113 (27.2) | 135 (32.2) |
| HIV status disclosure | Does not know status | 87 (21.8) | 71 (18.3) |
| | Knows status, has not disclosed | 177 (44.3) | 196 (50.5) |
| | Knows status, has disclosed | 136 (34.0) | 121 (31.2) |
| | Missing | 21 | 33 |
| Orphan (*2 missing*) | Both parents alive | 207 (49.3) | 215 (51.2) |
| | Single orphan | 136 (32.4) | 109 (26.0) |
| | Double orphan | 77 (18.3) | 96 (22.9) |
| Viral load (*9 missing*) | ≥1,000 copies | 151 (36.4) | 141 (33.7) |
| | <1,000 copies | 264 (63.6) | 277 (66.3) |
| SSQ score | No red flag | 265 (63.0) | 282 (67.0) |
| | Red flag | 156 (37.1) | 139 (33.0) |
| SSQ score | Median (IQR) | 8 (7–9) | 8 (7–10) |
| PHQ-9 score (*1 missing*) | Minimal (0–4) | 81 (19.3) | 82 (19.5) |
| | Mild (5–9) | 201 (47.9) | 181 (43.0) |
| | Moderate (10–14) | 106 (25.2) | 116 (27.6) |
| | Moderately severe/Severe (15–27) | 32 (7.6) | 42 (10.0) |
| PHQ-9 score | Median (IQR) | 8 (5–11) | 8 (5–11) |
| EQ5D index score | Median (IQR) *1 = best* | 0.84 (0.74–1) | 0.81 (0.74–1) |

IQR, inter-quartile range; PHQ, Patient Health Questionnaire; PST, problem-solving therapy; SSQ, Shona Symptom Questionnaire.

average, Zvandiri-PST participants received more text messages (mean difference = 6.50; 95% CI 0.57, 12.43, $p = 0.03$) and fewer support group sessions (mean difference = −1.88; 95% CI −3.51, −0.25), $p = 0.02$) than Zvandiri arm participants, with no evidence of a difference between arms in the mean number of facility visits, outreach visits, or contacts with caregivers.

According to self-reported data, 74 participants switched ART regimen during follow-up (11.7% of 632 who knew their status and were asked the question). The most common reported reason for switching regimen was that viral load was high ($n = 32$, 43.2%). In the Zvandiri-PST arm, 43 (13.9%) reported switching regimen compared to 31 (10.2%) in the Zvandiri arm.

## Qualitative findings

The 2 FGDs were held during the endline data collection period (January 2020). Each comprised 10 CATS and lasted approximately 90 minutes. Of the 240 planned case reviews (20 CATS have one per month for a year), 200 were conducted. Case report transcripts generally ranged from 500 to 1,000 words.

**Table 3. Intervention effect on primary and secondary outcomes at 48 weeks.**

| | Zvandiri-PST | Zvandiri | | | |
|---|---|---|---|---|---|
| **Binary** | **n/N (%)** | **n/N (%)** | **AOR (95% CI)** | ***p*-value** | **ICC** |
| *Primary outcome* | | | | | |
| Viral load ≥1,000 | 55/375 (14.7) | 45/379 (11.9) | 1.29 (0.68, 2.48) | 0.44 | 0.15 |
| *Secondary outcomes* | | | | | |
| SSQ≥ 8 | 9/377 (2.4) | 40/388 (10.3) | 0.19 (0.08, 0.46) | <0.001 | 0.17 |
| PHQ-9 ≥10 | 11/377 (2.9) | 34/388 (8.8) | 0.32 (0.14, 0.78) | 0.01 | 0.18 |
| EQ-5D index score <1 | 104/377 (27.6) | 151/388 (38.9) | 0.56 (0.31, 1.03) | 0.06 | 0.22 |
| **Continuous** | **Mean (SD)** | **Mean (SD)** | **AMD (95% CI)** | ***p*-value** | |
| SSQ score | 2.22 (2.15) | 3.38 (3.02) | −1.14 (−1.80, −0.49) | 0.001 | 0.18 |
| PHQ-9 score | 2.40 (3.01) | 3.48 (3.83) | -1.14 (−2.01, −0.27) | 0.01 | 0.19 |

AMD, adjusted mean difference; AOR, adjusted odds ratio; ICC, intracluster correlation; PHQ, Patient Health Questionnaire; SSQ, Shona Symptom Questionnaire.

All analysis adjusting for baseline value of the outcome, baseline age, and clinic as a random effect.

Trial participants and CATS reported that the PST intervention was difficult to implement, with challenges reflecting the relational context of adolescents' lives. Their problems tend to be a product of how entangled and dependent they are on their relationships with others, which limits the agency that youth have to resolve their problems themselves. Consequently, there was low fidelity to the later steps in the model, which involved developing SMART action plans to address an identified problem. Instead, the CATS focused on the earlier steps of the model (problem discussion), which participants found to be affirming and constructive in enabling them to cope.

The analysis is presented through 3 key themes: fidelity impeded by relational nature of adolescents' problems; the value of peer-led problem discussion therapy; and role for others in developing mental health support networks (Table 4).

## Fidelity impeded by relational nature of adolescents' problems

The CATS were able to encourage participants to discuss their problems, although it often took several sessions for them to feel confident to talk openly. This reticence indicates how unusual it is for adolescents to have a dedicated forum to talk about their problems, as well as the time taken to establish the expertise of the CATS in the eyes of the participants and their caregivers. This reflects a prevailing view articulated by adolescents, caregivers, and the CATS

**Table 4. Description of care received by arm.**

| | Zvandiri-PST, mean (SD), range | Zvandiri arm, mean (SD), range | MD (95% CI) | ***p*-value** |
|---|---|---|---|---|
| PST sessions | 5.0 (1.7), 0–11 | 0 | | |
| CKT sessions | 4.1 (2.6), 0–20 | 0 | | |
| Facility visits | 10.6 (3.9), 1–23 | 9.8 (4.4), 1–25 | 0.94 (−0.75, 2.62) | 0.28 |
| Text messages | 21.9 (16.0), 0–107 | 15.2 (10.3), 2–63 | 6.50 (0.57, 12.43) | 0.03 |
| Support groups | 7.1 (3.7), 0–17 | 8.9 (3.7), 0–16 | −1.88 (−3.51, −0.25) | 0.02 |
| Outreach visits | 7.5 (3.9), 0–25 | 7.0 (7.5), 0–120 | 0.50 (−1.58, 2.59) | 0.64 |
| Caregiver contact | 10.8 (4.9), 0–29 | 10.4 (6.1), 2–36 | 0.32 (−2.15, 2.79) | 0.80 |

CKT, Circle Kubatana Tose support group; MD, mean difference; PST, problem-solving therapy.

Adjusting for clinic as a random effect.

themselves that young people exercise very limited influence over their situations if they do not have adult support (Table 5, Subtheme 1a). As a CATS describes:

> *"Some participants look at my age and they tend to judge me and think I know nothing, or I am too young to help solve their problems. So, at times they don't open up to me about all their challenges."—Chiredzi*

The steps of the model, which involved identifying an action plan to address a particular problem, were less evenly implemented. Younger participants particularly struggled with the tasks of brainstorming problems and solutions within the time frame of each session. However, the primary factor that constrained the CATS was a recognition, borne out of their discussions with participants, which the adolescents had relatively little capacity to resolve their problems.

> *"She couldn't make a decision on which problem to tackle first: that of her parents fighting, or failure to afford school fees. She could not solve either of these problems by herself. So, it was extremely difficult for her to even choose the problem to deal with. It took her some time to finally speak up regarding the problem that she wanted to work on."—Murewa)*

Consistently, the problems identified by participants were embedded within their relationships (e.g., bereavement or abuse) or related to structural conditions of poverty. Both the drivers and solutions to their problems were inextricably connected to someone else, usually a significant adult in their lives. The participants increasingly trusted the influence and advice of the CATS, but neither the participants nor the CATS expected that the problems could be resolved without the intervention of significant adults. A CATS described a common challenge: *"There are some problems that were not within the participant's ability to solve"—Chiredzi*

When participants developed a SMART plan, they remained reliant on others to implement solutions. It was common for these plans to fail due to a lack of engagement or capacity from those they depended upon to instigate such actions (Table 5, Subtheme 1b).

> *"He told them [grandmother and maternal aunties] but they did not take any action. He continued to be sent home (from school due to no fees). The caregivers are refusing to look for money to pay for his school fees."—Gokwe South*

The emphasis on identifying solutions within the model was ill-suited to many adolescents' relational context. The CATS had limited experience to draw on to guide the participants in identifying feasible solutions. Without further support to develop and instigate an effective action plan the experience could at times be intimidating and disempowering. The dissonance between what they were expected to achieve in these sessions and what they felt was feasible caused some distress to participants. The CATS, who tended to be dedicated practitioners and took great pride in accomplishing their roles, were keen to emphasise that their limited capacity to implement all the steps in the model should not be interpreted as a failure to correctly engage.

> *"Some [problems] due to the situation they cannot be dealt with. For example [a client] thought if she sees where her mother was buried it can help her in her life but because of a lack of resources she could not go and see where her mother was buried."—Kwekwe*

**Table 5. Qualitative themes and indicative quotes on fidelity of delivering the intervention.**

| Themes | Subthemes | Indicative quotes |
|---|---|---|
| **1. Impeded fidelity** | | |
| | **1a. Scepticism of the capacity of young people to affect their peers' problems** | |
| | | *"It's still very difficult because some participants are still failing to open up because they will be thinking that how will I help them because I am also a child myself so it will be useless for them to share with me their problems." (Matobo)* <br> *"Some of the challenges are that some adolescents do not cooperate, they just look at you to the extent that it becomes very difficult to assist them. It is still very difficult to see whether PST is working or not. There is a problem of not understanding each other, it is difficult for a child to tell you their problem. Other participants think that it is very difficult to disclose their problems to another adolescent and still think they might not get assistance from the CATS."—Matobo* <br> *"Caregivers now understood and accepted us. At times you would reach a point when we would be chased by caregivers and you would actually see that it was difficult to get to the caregivers. But with time we were being treated nicely and welcomed. Caregivers allowed us to meet their children." (Gokwe South)* <br> *"The first day we arrived with aunty the child tried to run away and we saw the child after a long time. After we had waited at the child's home and then aunty talked to the child and it helped the child. When l visited the child for the second, third time alone we were now getting along well."—Kwekwe* |
| | **1b: Limited agency of youth to resolve relationally entangled problems** | |
| | | *"His father married a wife who was his stepmother and this was another challenge the client faced. He says there was never a good relationship between the two. The client was unhappy at home because the stepmother was not supportive of him."—Beitbridge* <br> *"The problem she mentioned is that her stepmother punishes her by depriving her food even for little offences and giving her punishments. The help that is required is to find someone, a mature and respectable person to have a diplomatic chat with her stepmother such that she can be enlightened on the need for her child to have adequate food and not be overburdened with work. So with such an intervention her situation can improve."—Zaka* <br> *"She won't be able to solve them because she is still very young. As a child she cannot do anything about her parents' conflicts and fights. When it comes to paying her own school fees she can't do that because she cannot be employed at her young age."—Murewa* <br> *"The SMART action did not work, the father was willing to assist but the mother refused."—Gwanda* <br> *"She mentioned that she was always insulted by her grandmother over the death of her parents, which then leads her to be depressed to such an extent that she does not take her medication. . . . . . .Her goal was not to be insulted by her grandmother over the death of her parents... . . It had no time frame as she wanted to talk to her grandmother's friend first because she was the one who would then decide when she will be able to talk to her grandmother but she managed to do that the Sunday after we met.. . . . . . .The plan did not work out because the grandmother's friend refused to talk to the grandmother claiming that she wouldn't understand her. . . . . . . The challenge that I faced was that the participant was not able to air out her thoughts, and spend most of the time crying."—Murewa* <br> *"His father married a wife who was his stepmother and this was another challenge the client faced. He says there was never a good relationship between the two. The client was unhappy at home because the stepmother was not supportive of him."—Beitbridge* <br> *"With the older ones you could talk to them and they would understand, but the younger ones did not understand and could not tell what was affecting them and you would give them a paper to draw."—Beitbridge* <br> *"Problem identification is mostly useful when discussing with a mature person. This beneficiary was young, she was slow in brainstorming that sometimes she could stop and tell me that she needs to think it through then come back later."—Chivi* <br> *"These [problem solving] steps are helpful in some scenarios but in others they are a bit difficult. For example about this beneficiary, she told me that she can go and look for firewood but looking at her age it was not possible for her to look for firewood to sell and be able to buy what she needed."—Zaka* <br> *"The problem she mentioned is that her stepmother punishes her by depriving her food even for little offences and giving her punishments. The help that is required is to find someone, a mature and respectable person to have a diplomatic chat with her stepmother such that she can be enlightened on the need for her child to have adequate food and not be overburdened with work. So with such an intervention her situation can improve."—Zaka* <br> *"The SMART action did not work, the father was willing to assist but the mother refused."—Gwanda* |
| **2. Value of peer-led problem discussion therapy** | | |
| | | *"I think [talking about problems] is helping because if the children have problems at home they are afraid to discuss with the adults but it is very easy when they share their problems with their peers."—Matobo* <br> *"The child is now able to open up when talking even when he meet others he is now able to open up and share challenges which will be troubling them. I can see that this has changed even his life at school for the better, he was stigmatising himself considering himself as a sick person but this changed."—Chivi* <br> *"I feel elated because even those who used to wear weary faces, they are now wearing happy faces. They are now happy. . . there is a great improvement since we can now freely talk and interact with each other which are a thing that was not possible before."—Murewa* |
| **3. Including others in developing broader mental health support networks** | | |

*(Continued)*

**Table 5.** (Continued)

| Themes | Subthemes | Indicative quotes |
|---|---|---|
| | **3a. Support for clients from significant adults** | |
| | | *"The problem is difficult and the solution might not work. It was a family issue so I could not directly involve myself."—Murewa*<br>*"I need an older person who can help facilitating the disputes between the participant's mother and the aunt."—Chiredzi*<br>*"I think if possible there can be a friend of the beneficiary to whom she is free to talk to or any close relative whom she can openly disclose what is troubling her, if such a people could be present during the session l think this could help."—Chivi*<br>*"I think if we can make his/her caregiver be available during the PST sessions this may help because he/she sometimes may not speak out solutions because he/she is young."—Gokwe South*<br>*"As CATS we want help so that parents are communicated with so that beneficiaries will be able to communicate with them."—Kwekwe* |
| | **3b. Support for CATS** | |
| | | *"When discussing with them about their problems it is difficult at times but as a CATS I am not supposed to be overwhelmed by it as I am supposed to help the beneficiaries in their various problems."*<br>*"The challenges are that when they discuss their problems with me they expect me to solve their problems or maybe give them that thing that they are in need of e.g. money to buy what they need."—Zaka*<br>*"It was difficult conducting counselling with young mothers because some of them talked about deep issues such as marriage life."—Zaka*<br>*"The main challenge that we have is that it is very difficult for a client to just open up because they judge you not knowing that you might be facing greater challenges, so you end up discussing your problems with them so that they are also free to open up their problems with you."—Matobo*<br>*"The beneficiary is requiring me to give him solutions to solve his problem, yet we encourage that the child should chose his/her own solutions to solve his problem."—Chivi*<br>*"I need you [the supervisor] to help me with ideas and skills on how to deal with very complicated problems during PST sessions."—Chiredzi*<br>*"The assistance that I need as a CATS is that the nurses should assist me in identifying problems since they are adults some children might be able to open up their problems with them; that is my wish."—Murewa*<br>*"I feel happy and that l have assisted someone, even though sometimes l feel unstable due to the problems that are shared."—Kwekwe* |

CATS, Community Adolescent Treatment Supporters; PST, problem-solving therapy; SMART, specific, measurable, achievable, realistic, and timely.

### Value of peer-led problem discussion therapy

Despite these challenges, most CATS considered that the intervention had been valuable. The primary benefit they described was that it had created a rare forum for adolescents to legitimately discuss their problems, beyond being asked about HIV and their treatment adherence, and to be listened to with empathy by a trusted peer. The opportunity to have their problems recognised and validated helped to reduce the perceived harm of the problem. As one CATS put it, *"They no longer had to suffer alone."—Gokwe South.* This suggests that the intervention operated as a form of problem discussion therapy, which was beneficial even if the problems could not be resolved. For additional quotes, see Table 5, Theme 2.

> *"The community used to discriminate her, and through our sessions she understood that people are always saying something: good or bad she will take her medication."—Kwekwe*

### Including others in developing broader mental health support networks

The third key theme was the need to extend the mental health support network provided for the adolescents beyond the CATS. Given the uneven relational power characterising many of the problems encountered by participants, the CATS emphasised that it was necessary to include significant others, especially caregivers, in the intervention. This would not only

expedite the development of trust and rapport, but also strengthen caregivers' engagement in the action plans (Table 5, Subtheme 3a).

> *"Challenges encountered by the beneficiary are beyond their ability to mention or answer to without a parent being present. You may find that the parent's fight is the reason behind the problem that the child is facing, the parent is the one whom we need to talk to before talking to the child so that we can help the child."*

Some CATS who included caregivers in their sessions described the benefits: *"It was difficult doing PST sessions only with children but when we were allowed to work with caregivers it now became easy."—Gokwe South*

Even with this potential avenue of support, many problems were also beyond the remit of the participant's immediate community to resolve. CATS described this as "overwhelming" at times, indicating the need for investment in robust referral systems to professional social welfare or psychological review:

> *"When we discuss bigger problems, we will be needing help from mentors and nurses at the hospital so that it will be easier for us as CATS in our discussions."—Zaka*

The CATS described the work as gruelling at times and highlighted the need for ongoing supervision from adult mentors to support them emotionally and professionally and protect them from potential occupational harm (Table 5, Subtheme 3b).

> *"I thought by this time I would say it is easy but each time I meet the participants they are always revealing something new. . . ... Supporting someone with a common mental disorder feels like you have to carry them around in order for them to cope with life."—Hwange*

## Discussion

Among adolescents with HIV and comorbid symptoms of CMD, peer mental health counselling showed no evidence of an impact on the primary outcome of virological nonsuppression but did show apparent evidence of improved mental health. To our knowledge, this is the first trial to show evidence of improved mental health in this population group. Those who received care from a PST-trained CATS had substantially lower prevalence of CMD and depression symptoms after 1 year, compared to those receiving care from a CATS without PST training. There was weak evidence of improved quality of life (EQ-5D). Mental health outcomes improved over time in both trial arms.

This trial adds to the sparse literature on mental health interventions for adolescents living with HIV in low- and middle-income settings (LMIC). A recent systematic review identified only 3 such studies [16]. Two were pilot studies that were not powered to assess effectiveness [17,18], and the third was an analysis of the baseline characteristics of trial participants [19].

The results of the current trial extend those of our previous trial, which was conducted among adolescents living with HIV and was not restricted to those with symptoms of CMD. In that trial, which compared the Zvandiri programme delivered by CATS (i.e., the control arm in the current trial) with standard HIV care, 20.4% of participants had an SSQ score ≥8 at enrolment. We found an intervention effect on virological nonsuppression or death but not on mental health outcomes [5]. Prevalence of virological nonsuppression or death was 24.9% in the Zvandiri arm versus 35.9% in the standard HIV care arm after 2 years (adjusted prevalence ratio = 0.58; 95% CI 0.36 to 0.94).

In our present trial, there was a substantial reduction in virological nonsuppression in both arms (33.7% to 11.9% in the Zvandiri arm, 36.4% to 14.7 in the Zvandiri-PST arm). There was no evidence of added benefit of PST on virological nonsuppression. There may not be scope for further reduction of virological nonsuppression below the low level achieved with Zvandiri care, particularly given that some nonsuppression is due to ART resistance rather than suboptimal adherence. Evidence of the extent of ART resistance in Zimbabwe is limited, but there are reasons to believe it is particularly high among adolescents. A study of 102 children and adolescents in Harare with virological failure in 2012 found that 68% of them had at least one clinically significant mutation [20].

This model integrates mental healthcare for adolescents into HIV care, as recommended in a recent review [21] and identified as a priority area by the Adolescent HIV Implementation Science Alliance [22]. Our finding that PST, as originally envisaged, was not appropriate or feasible for an adolescent population is in line with the literature. Adolescents have very limited agency to resolve their own problems and rely on others to help them [23]. An important element of PST is for the counsellor to help the client choose a problem that is both meaningful and within their power to change [9]. The extent of challenges that adolescents faced may have made it less feasible to help them to focus on "smaller" problems that were in their control. In our study, younger adolescents became tired over the course of a session and also had trouble brainstorming solutions within the time. The intervention was equally effective on mental health outcomes for younger adolescents as older ones, but this could be because the CATS adapted it towards "problem discussion therapy."

Caregivers have a role to play in adolescent mental health and HIV interventions [24]. In our trial, CATS pointed out that the minimal involvement of caregivers limited the capacity of the adolescent and CATS to resolve problems. The ZENITH trial in Zimbabwe showed improvement in HIV virological suppression of children aged 6 to 15, from a family-centred home-based support programme by lay workers [25,26].

This study supports previous evidence that young people gain fulfilment from being CATS [27], but it can also be challenging and emotionally draining, particularly when CATS are faced with problems that they do not have the resources or experience to handle. The needs of young peer supporters have been laid out in the TRUST framework [23], comprising Training, Referral pathways, Understanding the remit of their role, Supervision, and recognition that Talking helps.

The strengths of the trial are that it was well powered with low intracluster correlation, the sample was representative of the whole country [7], follow-up was good (90.9% overall), and the qualitative evaluation enabled correct interpretation of the quantitative results. Limitations included that cost data were not available and that the mental health tools were validated in adult populations, not adolescents. The ways that the intervention was adapted may limit external application of the findings.

## Conclusions

Virological nonsuppression is common among Zimbabwean adolescents with mental health problems living with HIV. This trial provides additional supporting evidence of the effect of Zvandiri on virological suppression. Training in counselling for peer supporters can have beneficial effects on mental health. We recommend that PST for adolescents should be less structured than the SMART plan for adults, with shorter sessions (especially for young adolescents) over a longer time frame. Counsellors should be trained to help clients identify problems that are within their power to affect. Creating a space for discussion and the sharing and discussion of problems, rather than solving them, should also be valued. Finally, ways to involve caregivers, such as family therapy approaches, should be explored.

## Supporting information

**S1 CONSORT Checklist.**
(DOCX)

**S1 Text. Trial analytical plan.**
(PDF)

**S1 Table. Association of baseline characteristics with missing primary outcome.**
(DOCX)

**S2 Table. Unadjusted intervention effect on primary and secondary outcomes at 48 weeks.**
(DOCX)

**S3 Table. Effect modification of baseline age on trial outcomes at 48 weeks.**
(DOCX)

## Author Contributions

**Conceptualization:** Helen A. Weiss, Dixon Chibanda, Nicola Willis.

**Data curation:** Victoria Simms, Abigail Mutsinze, Carol Wogrin.

**Formal analysis:** Victoria Simms, Helen A. Weiss, Sarah Bernays, Carol Wogrin.

**Funding acquisition:** Helen A. Weiss, Dixon Chibanda, Nicola Willis.

**Investigation:** Silindweyinkosi Chinoda, Abigail Mutsinze, Sarah Bernays, Ruth Verhey, Carol Wogrin.

**Methodology:** Helen A. Weiss, Sarah Bernays, Dixon Chibanda.

**Project administration:** Silindweyinkosi Chinoda, Abigail Mutsinze, Tsitsi Apollo, Owen Mugurungi, Dorcas Sithole, Nicola Willis.

**Resources:** Tsitsi Apollo, Owen Mugurungi, Dorcas Sithole, Dixon Chibanda, Nicola Willis.

**Supervision:** Helen A. Weiss, Ruth Verhey, Carol Wogrin.

**Visualization:** Victoria Simms.

**Writing – original draft:** Victoria Simms.

**Writing – review & editing:** Helen A. Weiss, Silindweyinkosi Chinoda, Abigail Mutsinze, Sarah Bernays, Ruth Verhey, Carol Wogrin, Tsitsi Apollo, Owen Mugurungi, Dorcas Sithole, Dixon Chibanda, Nicola Willis.

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
