## [Editor Report · Decision Letter 0]

28 Jun 2021

Dear Dr Simms, 

Thank you for submitting your manuscript entitled "Effect of peer-led counselling on viral load and mental health of adolescents living with HIV in Zimbabwe: a cluster-randomised trial" for consideration by PLOS Medicine.

Your manuscript has now been evaluated by the PLOS Medicine editorial staff and I am writing to let you know that we would like to send your submission out for external assessment.

However, we need you to complete your submission by providing the metadata that is required for full assessment. To this end, please login to Editorial Manager where you will find the paper in the 'Submissions Needing Revisions' folder on your homepage. Please click 'Revise Submission' from the Action Links and complete all additional questions in the submission questionnaire.

Please re-submit your manuscript within two working days, i.e. by Jun 30 2021 11:59PM.

Once your full submission is complete, your paper will undergo a series of checks in preparation for external assessment. 

Kind regards,

Richard Turner, PhD

rturner@plos.org

---

## [Decision Letter · Decision Letter 1]

7 Aug 2021

Dear Dr. Simms,

Thank you very much for submitting your manuscript "Effect of peer-led counselling on viral load and mental health of adolescents living with HIV in Zimbabwe: a cluster-randomised trial" (PMEDICINE-D-21-02706R1) for consideration at PLOS Medicine. 

Your paper was discussed among the editors and sent to independent reviewers, including a statistical reviewer. The reviews are appended at the bottom of this email and any accompanying reviewer attachments can be seen via the link below:

[LINK]

In light of these reviews, we will not be able to accept the manuscript for publication in the journal in its current form, but we would like to invite you to submit a revised version that addresses the reviewers' and editors' comments fully. You will appreciate that we cannot make a decision about publication until we have seen the revised manuscript and your response, and we expect to seek re-review by one or more of the reviewers. 

We hope to receive your revised manuscript by Aug 30 2021 11:59PM. Please email us (plosmedicine@plos.org) if you have any questions or concerns.

Please let me know if you have any questions, and we look forward to receiving your revised manuscript. 

Sincerely,

Richard Turner PhD

Senior editor, PLOS Medicine

rturner@plos.org

Please quote aggregate demographic details for study participants in the abstract.

Please quote study dates in the abstract.

Please add a new final sentence to the "Methods and findings" subsection of your abstract, beginning "Study limitations include ..." or similar and quoting 2-3 of the study's main limitations. 

At line 52 and elsewhere in the elements of discussion in the paper, please quote the null findings on the study's primary endpoint prior to mentioning other findings.

After the abstract, please add a new and accessible "Author summary" section in non-identical prose. You may find it helpful to consult one or two recent research papers in PLOS Medicine to get a sense of the preferred style. 

Please avoid claims such as "the first", e.g., at line 339, and where needed add "to our knowledge" or similar.

Please remove the information on funding from the end of the main text. In the event of publication, this will appear in the article metadata via entries in the submission form. 

Throughout the text, please adapt reference call-outs to the following style (noting the absence of spaces within the square brackets): "... symptoms of CMD [10,11].".

Noting reference 7, please ensure that all citations contain full access information. 

Please use the journal name abbreviation "PLoS ONE" in the reference list. 

Please rename figure 1 "Participant flowchart" or similar. 

Please include a completed checklist for the most appropriate reporting guideline, e.g., CONSORT, with your revision as an attachment, labelled "S1_CONSORT_Checklist" or similar and referred to as such in the Methods section. 

In the checklist, please refer to individual items by section (e.g., "Methods") and paragraph number, not by line or page numbers as these generally change in the event of publication. 

Comments from the reviewers:

*** Reviewer #1: 

Statistical review

This paper reports a cluster randomised trial comparing a peer-led counselling intervention on HIV-positive adolescents with mental health problems.

Generally the results are reported well. I have some minor comments on the statistical methods and reporting, which are listed below.

1. Abstract conclusions - I would recommend flipping the sentence around to emphasise the primary outcome was not met but that there were some significant secondary outcomes. The same goes for the first paragraph in the discussion.

2. Methods page 5 - can more be said about the randomisation procedure, was it a block randomisation or was there any stratification?

3. Methods - I'd recommend having the outcomes listed first and then the sample size section.

4. Page 6 - the primary outcome here is given as 1000 copies per microlitre, whereas the protocol says 100 copies per millilitre - these are different by a couple of orders of magnitude - can the authors clarify? The secondary outcomes also have slightly different thresholds to those given in the protocol.

5. Results, page 9 - for the subgroup analysis I would mention the significant interaction p-value as otherwise it might be puzzling to the reader how the two non-significant subgroup p-values are evidence of a significance. 

James Wason

*** Reviewer #2: 

This is an excellent paper, and an important study. It compares a well-evidenced NGO program to support adolescents living with HIV, to an enhanced version with additional mental health components based on another well-evidence-based program. It is encouraging to see a commitment to building the evidence-base and optimising interventions, and to see two research groups working together (which doesn't happen as often as it should). 

The original Zvandiri program has strong evidence of improving viral load suppression. The original Friendship Bench program has strong evidence of reducing common mental health disorders. This study shows that a combined program can do both, even with some complexities and challenges. This is of enormous value to programming for this group of exceptionally vulnerable adolescents.

The methods are robust, and the program delivery was done through existing government clinics, suggesting scaleability. Good measures are used. I had no comments on the methods or write-up, which are both excellent. I'm not familiar with a quadrature check for model fit, but the other analytical approaches all look strong. 

The qualitative findings were extremely valuable - and it would be helpful to know whether these have contributed to any adjustments to the mental health component of the program. In retrospect, it may have been useful to have done the qualitative evaluation before the trial, to allow adjustments to be made, but it is also absolutely commendable to have accompanied the quantitative trial with qualitative study, and in the wider implementation science approach these are all iteratively feeding into program improvements. 

I was struck by the importance of including caregivers, and at some point it could be of value to think about synergies between these approaches and the growing evidence-base on parenting programs for families of adolescents - which have not always shown good impacts on adolescent mental health, but have helped with some of the structural challenges mentioned, such as family budgeting, violence prevention, reducing conflict and improving family relationships.

My only minor comments are: 

1. Exclusion criteria included intoxication or psychosis. There is a lot of logic to this, but in most of the countries most affected by HIV, there would be very limited other services for these adolescents, and so it would be valuable also to see whether the combined program would help even in quite extreme cases. 

2. It would be helpful to have a bit more information about the delivery of the combined program.

3. I know that there is a lot described in the protocol paper, but perhaps some more information could be uploaded into the supplementary material to this paper, so that it is easy for the reader to access. 

But these are very minor. This is clearly a paper of real importance to the field, and a novel and extremely important study. Congratulations to the team. 

Lucie Cluver. 

*** Reviewer #3: 

PLOS Medicine review

The authors conducted a clinic-randomized trial (n=60 clinics) to assess the impact of adding "training in Problem-Solving Therapy (PST)" to an established peer counselling intervention on mental health and viral suppression outcomes among adolescents with HIV.

Mental health is a predictor of adherence and retention in HIV care as well as an important health outcome in its own right. Adolescents with HIV face unique mental health challenges and this population is growing as vertically-infected children age into adolescence (e.g. Maskew et al. 2019, Lancet HIV).

Whereas the (standard of care) established counselling intervention was previously shown to be effective in increasing viral suppression, it had no impact on mental health. Hence the effort to amend the intervention with PST.

PST appears to be the missing link. Although the authors find no impact of the PST-augmentation on viral suppression (perhaps due to ceiling effects), they find large reductions in depression and other adverse mental health outcomes. (One can infer that relative to "no counselling", the combination of PST with the original intervention would therefore be expected to lead to improvements on both mental health and viral suppression.) 

The study is important and appears carefully executed (with carefully described procedures). The combination of qualitative data is a major asset in interpreting the study findings, although I think it can be better used (see below).

General comments: 

1) RATIONALE. I think the study would be strengthened by a clearer explication of the intervention rationale (why PST?). The content of PST-training (Friendship Bench) is described. However, what are the theoretical constructs that PST is designed to address? And how did the authors identify these constructs (based on their prior work) as needing to be addressed? My initial sense is that the intervention was designed to (a) give people a sense of agency and control in their lives and (b) to provide cognitive tools to assist with problem solving and give opportunities for practice using these tools. I infer that a major source of mental morbidity is a sense of lack of control which may be exacerbated by having HIV. And building up a sense of agency as well as problem solving skills could theoretically support ART adherence too. (But these were my own speculations based on the paper. Further details here would help the reader understand the nature of the problem that needs to be solved. In fact, the bottom of p10 suggests a different interpretation based on coping and validation rather than agency.)

2) FIDELITY, FEASIBILITY, and ACCEPTABILITY. There is a bit of whiplash as the authors present large intervention effects and then present qualitative data indicating that the intervention was actually rather difficult to implement (low feasibility, acceptability) and had low fidelity in implementation. So, which is it? If the data indicate that the intervention was not really implemented faithfully, should we interpret this as evidence that the highly significant effects on mental health occurred by chance? Or should we rather interpret this as evidence that the particular components of the intervention that were implemented turned out to be important in this context? 

I love the combination of qual and quant. But as it is, my feeling is that the qual results undercut the quant findings, rather than adding validity and nuance. The qual results are framed as providing information on "what was wrong about the intervention implementation." But this is immaterial. The question in the reader's mind after the quantitative results is "what was RIGHT about the intervention implementation", i.e. what led to the big mental health impacts observed?

I think the discussion of fidelity would be greatly improved if it were organized around key questions that guide interpretation of the quantitative results. E.g., (a) can the authors provide some evidence that study participants in the treatment group actually received something different than participants in the control group? (b) can the authors be more explicit about WHAT exactly was different? E.g. if it seems that the earlier stages of PST were emphasized, then that's valuable to know. (c) How do the findings on fidelity shape interpretation of the theoretical model? i.e. if the intervention was designed to affirm the experiences of subjects, to build locus of control, and to build cognitive problem-solving skills… but if only the first of these was actually implemented, then that is pretty important for interpretation. But the authors should set this up ex ante that the intervention was designed to target different constructs. And (d) how can this evidence guide future development of the intervention and inform scale-up?

Smaller stuff:

ABSTRACT

- I found the intervention content and rationale to be unclear. The abstract states that the comparison was between peer-counselling and peer-counselling where the counsellor was trained in Problem-Solving Therapy. However, it would be helpful to know: (a) how did PST differ from the standard of care counselling; (b) did the counsellors actually deliver PST or were they just trained in PST; (c) what was the theoretical motivation for PST relative to standard of care, i.e. what was the problem that PST was introduced to solve?

- In the description of results, it is unclear what 14.7% and 11.9% refer to. Are these prevalences of viral non-suppression? Or percent reductions in non-suppression? Or percentage point reductions in non-suppression?

- Conclusion "Active involvement of caregivers and strengthened referral structures could increase feasibility and effectiveness" doesn't seem to follow from the study results.

METHODS

- "An a-priori decision was made to adjust for baseline values of the relevant outcome measure and for key variables that were deemed imbalanced between arms at baseline or were associated with missing outcome data." The authors should also present crude/unadjusted results. And the decision rules for how variables were determined to be "key" and therefore adjusted for should be stated.

RESULTS

- "The proportion of participants with an EQ-5D score below 1 was slightly lower in the Zvandiri-PST arm than in the Zvandiri arm (27.6% vs 38.9%; AOR=0.56; 95%CI 0.31, 1.03; p=0.06)." The term "slightly lower" is inaccurate. The PST arm proportion was >40% lower than control! The p-value was "slightly higher" than conventional cut-offs for statistical significance (but the CI is all that matters anyway). 

CONCLUSIONS

- "Our finding that PST, as originally envisaged, was not appropriate or feasible for an adolescent population is in line with the literature." I don't understand. It feels like the authors are saying that the intervention didn't work. But it DID work! (unless I misunderstand)

Final word: 

The availability of both qualitative and quantitative (trial) data is a major asset to this study. However, while each component (quant and qual) was well done, the current manuscript does not integrate the two very well. It feels like a trial results paper on the success of an intervention, followed by a qual study on how the intervention failed. This contradiction led to a great deal of confusion in this reader as to whether the trial results were credible and missed opportunities to explicate the reasons for the large trial effects. A cluster-randomized trial is higher quality evidence (per GRADE standards) and should dictate the overall conclusions of the study. I strongly recommend the authors cut the qual section by 50% and use the qual section to answer very specific questions that will help the reader interpret what went well (leading to large effects) in the intervention and what components of the PST intervention to emphasize going forward if it is brought to scale.

***

[LINK]

---

## [Decision Letter · Decision Letter 2]

22 Oct 2021

Dear Dr. Simms,

Thank you very much for submitting your revised manuscript "Effect of peer-led counselling on viral load and mental health of adolescents living with HIV in Zimbabwe: a cluster-randomised trial" (PMEDICINE-D-21-02706R2) for consideration at PLOS Medicine. 

Your paper was re-seen by our referees and discussed among the editors. The reviews are appended at the bottom of this email and any accompanying reviewer attachments can be seen via the link below:

[LINK]

In light of these reviews, we will not be able to accept the manuscript for publication in the journal in its current form, however we would like to like to invite you to submit a further revised version responding fully to the reviewers' and editors' comments. We cannot make a decision about publication until we have seen the revised manuscript and your response, and we expect to seek re-review by one or more of the reviewers. 

We hope to receive your revised manuscript by Nov 12 2021 11:59PM. Please email us (plosmedicine@plos.org) if you have any questions or concerns.

Please let me know if you have any questions, and we look forward to receiving your revised manuscript. 

Sincerely,

Richard Turner PhD

Senior editor, PLOS Medicine

rturner@plos.org

Noting reviewer 3's comments, we ask you to explain how the modification of the intervention fits in with the conduct and findings of the trial - did this happen before or, as apparently the case, during the study, for example?

We reserve judgement about the suggested restructuring of the paper pending resolution of the previous question. 

In the response to referees you mention that the intervention delivered was "problem discussion therapy" rather than "problem-solving therapy" per se, and it seems that judicious changes need to be made in the text, for example around line 140, to reflect this. 

We ask you to adapt the title to better match journal style, and suggest: "Peer-led counselling with problem discussion therapy for adolescents living with HIV in Zimbabwe: A cluster-randomised trial".

In the abstract and elsewhere in the paper, please soften the language describing secondary outcome findings, e.g., "... evidence of an apparent effect"

In the author summary, please present the primary outcome findings before those of the secondary outcomes. 

Please avoid claims of "the first", e.g., at line 92, and where necessary add "to our knowledge" or similar. 

Please substitute "sex" for "gender" where appropriate, e.g., in table 2.

Please update reference 23.

Comments from the reviewers:

*** Reviewer #1: 

Thank you to the authors for addressing my previous comments. Regarding the primary endpoint cut-point, this is denoted as 1000 copies/mL on page 2 line 35 and 1000 copies/µl (which I understand as microlitre rather than millilitre) on page 8 line 170. Otherwise I am happy with the paper and have no issues to raise.

*** Reviewer #2: 

I am happy with the authors responses to the reviewer and editor comments. Recommend acceptance. Lucie Cluver. 

*** Reviewer #3: 

Thank you for the opportunity to re-review this paper. I appreciate the clarifications that the authors have made in response to my comments. However, several of the underlying issues remain unaddressed.

1) The authors have clarified that "The intervention as originally envisaged was not appropriate or feasible. It was modified in practice. The modified version worked."

In light of authors' feedback, it is not clear what the intervention was. The intervention is described in the abstract as "training peer counsellors in PST". Yet if PST was not delivered, then it is not at all clear that training counsellors in PST was the key element of the intervention. What WAS the intervention? And how would someone replicate this intervention in another setting? Inasmuch as research is about creating generalizable knowledge, the intervention needs to be well-defined in order for the reader to draw any conclusions about it. This is a major limitation of the study.

I still believe the study is interesting and important. But...

(A) The authors should change how they present the intervention. They might consider labelling the study a "pilot" of a new, situationally-adapted mental health counseling intervention based on PST principles and training. They may also specify that this was a community-engaged intervention development approach, with peer counselors providing ongoing feedback and shaping the design of the intervention as it was implemented. And that the goals of the intervention were to engage counsellors in an effort to design wraparound mental health services, in which they had discretion to develop their own protocols. 

(B) The authors should invert the paper, describing the intervention development process, process outcomes, and intervention content (as implemented) first, BEFORE showing the data on impact of the intervention on the primary and secondary outcomes. As currently organized, it was hard for the reader to interpret the results without a clear understanding of the intervention first.

(C) Relatedly, the role of the qualitative data collection and analysis in the overall project needs to be specified. (I.e. the mixed methods design) Inasmuch as the qualitative research guided intervention development, it should be labeled as "formative research" and presented before the data on the trial results. Alternatively, if it was "explanatory research" of process outcomes, then it should be labeled as such and STILL presented before the other outcomes.

2) The authors did not respond to my request for further information on the theoretical constructs that the friendship bench intervention was designed to target, why these were relevant to the adolescent HIV population, and which of these constructs the ADAPTED version of the intervention (i.e. not PST) still targeted. Given that there was poor fidelity to the planned intervention, understanding what constructs were actually targeted successfully and which were not is critical for any kind of generalizability to other settings. A lot of this comes out in the qualitative results. But it would be advantageous to spell out at the beginning of the paper / in the methods section what constructs were being targeted.

3) The authors did not adequately address my comment #6. I requested crude/unadjusted results from the RCT. These are always appropriate whether pre-specified or not. Further, I requested information on how the authors determined what variables to include in the adjusted model. In the reply, it is unclear whether in fact there were pre-specified decision rules or not. They write "there were no specific decision rules" and then "the decision was made prior to analysis". If there was a pre-analysis protocol defining what was adjusted for, the authors can link to that. But if they chose variables as they went along based on their associations with the trt and outcome non-response, then they should show robustness checks with other adjustment variables. Basically, they need to convince the reader that they did not "choose" their preferred specification and that the benefits of randomization were preserved.

I remain convinced that this is an important study. However, I cannot recommend it for publication in its current form based on the methodological concerns highlighted above.

***

[LINK]

---

## [Decision Letter · Decision Letter 3]

28 Nov 2021

Dear Dr. Simms,

Thank you very much for re-submitting your manuscript "Peer-led counselling with problem discussion therapy for adolescents living with HIV in Zimbabwe: a cluster-randomised trial" (PMEDICINE-D-21-02706R3) for consideration at PLOS Medicine.

I have discussed the paper with editorial colleagues and it was also seen again by one reviewer. I am pleased to tell you that, provided the remaining editorial and production issues are fully dealt with, we expect to be able to accept the paper for publication in the journal.

[LINK]

Please let me know if you have any questions, and we look forward to receiving the revised manuscript.   

Sincerely,

Richard Turner PhD 

rturner@plos.org

Requests from Editors:

In your data statement (submission form), can you substitute a specific webpage address for data from the present study?

Is there a duplicate parenthesis at line 48 (abstract)?

At line 52, please adapt "was beneficial" to indicate how this was judged.

At line 56, please avoid "further reduction" (in favour of "... did not add to the benefit of peer support in reducing virological suppression" or similar). 

Please remove the duplicated full point at line 142. 

At line 257, please revisit "anadjusted".

Please add a further sentence, say, to the section on limitations (line 447). For example, perhaps the adaptation of the intervention might limit external application. 

Comments from Reviewers:

*** Reviewer #3: 

I thank the authors for their thoughtful responses and clarifications. I appreciate the addition of the crude RCT results. And I think the authors' justifications regarding manuscript organization are reasonable.

A final suggestion: to avoid the feeling of whiplash that I experienced in reading this article the first time, I would urge the authors to assist the reader a bit by defining the intervention a bit more broadly than PST, e.g. "discussion based therapy by peer counsellors trained in PST", and foreshadowing that there was no further intervention to ensure that peer counselors were staying close to PST protocols and that the counselors were free to use and interpret their training as they saw fit. ...or something like that. So that when you get to the process outcomes, the focus is on "why did it work" rather than on "looks like it didn't really work".

I believe it's a strong paper that will push the field forward.

***

[LINK]

---

## [Editor Report · Decision Letter 4]

9 Dec 2021

Dear Dr Simms, 

On behalf of my colleagues and the Academic Editor, Dr Bor, I am pleased to inform you that we have agreed to publish your manuscript "Peer-led counselling with problem discussion therapy for adolescents living with HIV in Zimbabwe: a cluster-randomised trial" (PMEDICINE-D-21-02706R4) in PLOS Medicine.

Prior to final acceptance, please update the data URL in the submission form; and convert nested parentheses (abstract) into square brackets. 

PRESS

Sincerely, 

Richard Turner, PhD 

rturner@plos.org